# Exploring and understanding the scope and value of the Parkinson's nurse in the UK (The USP Project): a realist economic evaluation protocol

Sarah Brown [iD],[1] Sonia Michelle Dalkin [iD],[2] Angela Bate,[1] Russ Bradford,[3] Charlotte Allen,[3] Katie Brittain,[4] Amanda Clarke,[5] Annette Hand [iD] [1]

¹Nursing, Midwifery and Health, Northumbria University, Newcastle Upon Tyne, UK
²Faculty of Health and Life Sciences, Northumbria University, Newcastle Upon Tyne, UK
³Parkinson's Concierge, London, UK
⁴Department of Nursing, Midwifery and Health, Northumbria University, Newcastle Upon Tyne, UK
⁵Health and Life Sciences, Northumbria University, Newcastle Upon Tyne, UK

**Correspondence to**
Dr Annette Hand;
a.hand@northumbria.ac.uk

## ABSTRACT

**Introduction** There are multiple configurations of specialist nurses working in the field of Parkinson's. Parkinson's Nurse Specialists (PNSs) are recognised as playing a pivotal role; however, there is little published evidence to demonstrate their effectiveness. Further evidence is needed to establish which aspects of the PNSs provide the greatest benefit to people with Parkinson's and their families, and the cost-effectiveness of different models of care.

**Methods and analysis** Realist approaches explain how and why programmes work (or not) through striving to answer the question: what works, for whom and under what circumstances. This research uses a realist evaluation and aims to integrate an economic analysis within the realist framework. We refer to this as 'realist economic evaluation'. It comprises four phases: (1) developing resource-sensitive initial programme theories (IPTs) using surveys to gain a better understanding of the role and impact (costs and benefits) of the PNSs; (2) testing the IPTs through qualitative interviews and quantitative data analysis; (3) evaluating the cost and resource use implications alongside the benefits associated with the role of the PNSs and (4) iteratively refining the IPTs throughout the project. The IPTs will draw on both quantitative and qualitative data. The result of the study will be a series of refined programme theories, which will explain how specialist nurses work in the field of Parkinson's in the UK, what impact they have on people with Parkinson's and their families and carers, and at what cost.

**Ethics and dissemination** Northumbria University, the Health Research Authority and Health and Care Research Wales have approved this study. Key findings will be disseminated throughout the duration of the project online and through social media, and via annual and regional Parkinson's meetings and the Parkinson's UK Excellence Network. Academic dissemination will occur through publication and conference presentations.

## INTRODUCTION
### Rationale for evaluation

Parkinson's disease (PD) is the second most common neurodegenerative condition in the UK, affecting around 145 519 people.[1]

### Strengths and limitations of this study

► To our knowledge, this is the first study to put realist economic evaluation into practice. This is the first stakeholder-driven national study to explore in-depth the role and value of the Parkinson's Nurse Specialists from multiple perspectives, while also exploring resource use, costs and cost-effectiveness.
► Patient and public involvement (PPI) members have been involved in this research from the outset, through meaningful activity such as engaging in regular team meetings, recruiting other people with Parkinson's and carers to be involved, developing surveys and promoting the study.
► This is a large study with a large amount of data collection. We will likely have to prioritise the programme theories we study in-depth. This will be done in collaboration with PPI members.
► In this study, we want to integrate the realist and economic evaluation. As eluded to above, this is the first study to use a realist economic evaluation design, and, while there are standard practices with respect to data collection and analysis for realist evaluation and economic evaluation, there are none for realist economic evaluation. Thus, methods of data collection and analysis will evolve with the study. However, the novelty of the design and time constraints, may limit the extent to which full integration of realist and economic approaches can be achieved.

It is a complex, progressive condition, often resulting in motor impairments (eg, movement and mobility problems) and non-motor symptoms (eg, sleep problems, cognitive impairment, depression and constipation).[2] The National Institute for Health and Care Excellence (NICE) guideline on PD[3] recommends that people with PD should have regular access to: clinical monitoring and medication adjustment; a continuing point of contact for support, including home visits when appropriate; and reliable

information about clinical and social matters of concern. These resources can be provided by a Parkinson's Nurse Specialist (PNS); a role that was created in response to recommendations from previous research.[4]

The creation of the PNS role aimed to improve prognosis, through better education and support and to reduce the impact of PD on people and their families and carers. The role complemented other members of the multiprofessional team (MPT), however, it was identified that nursing interventions and priorities differ across the PD trajectory.[4] In 1992, the Parkinson's Disease Society developed a team of nurses with a special interest and training in PD to improve standards and services for PwP.[5] A task force for the Parkinson's Disease Society recommended that each PwP was assigned a key worker to co-ordinate care and recognise changes in the disease trajectory, in collaboration with MPT members.[6] Today, the PNS is still recognised as playing this pivotal role,[7] with around 385 PNSs working in primary, secondary and tertiary care settings in the UK. However, NICE provides no detail around their role or caseload, and the guideline on PD[3] only references one single evaluation study.[8] Furthermore, stakeholders often describe the positive value of the PNS but there is little published evidence to demonstrate their effectiveness.

### Role of the PNS

The PNS provides assistance with ongoing management and follow-up through: medication review; clinical leadership; help with postdiagnostic counselling; education support and advice for people with Parkinson's (PwP), carers and other staff; signposting to other services; and case management.[9] Many run clinics, undertake home visits, refer on and co-ordinate care packages.[10] PNSs are often the first point of contact for PwP, ensuring fast access to specialist care, while relieving pressure on neurologist/geriatricians with a special interest in PD.[9]

### PNS education and training

Parkinson's UK offers an induction training programme for new PNSs. In 2018, Parkinson's UK launched the new learning pathway for PNSs to signpost and suggest areas of further education/development and to support newly appointed PNSs and more experienced PNSs. The Parkinson's Competency document,[11] describes the knowledge and skills required by the PNS to manage the care of PwP across healthcare settings. Over recent years, Neurology Nurses have been introduced in some areas to support PwP in what was traditionally a PNS role. Neurology Nurses care for people with a variety of neurological conditions in all healthcare settings.[9] However, it is unclear how many Neurology Nurses are currently caring for PwP in the UK, what training they have had and what services they provide. It is also unknown whether Neurology Nurses follow The Parkinson's Competency document[11] to demonstrate the expected knowledge and skills of a specialist nurse working in the field of Parkinson's.

### Evaluation of the role of the PNS

PwP living in the community supported by a PNS had improved subjective well-being at no extra cost, compared with those who were supported by GPs.[8] Compared with neurologists, PNSs provided longer consultations and paid more attention to patients' concerns.[5] No differences have been found between the two job roles in terms of health outcomes for PwP, although Reynolds *et al*[5] recognised the benefit of the two professions collaborating. In 2006, Eighty-nine PNSs were surveyed to examine job specification, perceptions of service delivery and views about assistance[12]: 80.9% of respondents had completed specialist training, 32.6% were prescribers and 60.7% had been in post for more than 5 years. The major barriers to service delivery were lack of time, lack of clerical help and heavy caseload. The study concluded that PNSs provided high quality, disease-specific care to PwP; however, nurses were concerned about the ability to maintain care standards. Another evaluation of the PNS role examined the perceived effectiveness, acceptability and efficacy among PwP, their carers and the MPT.[7] The highest satisfaction rating for a PwP was being able to contact the PNS if they developed side-effects from treatment, while lowest satisfaction was experienced when the PNS was unable to provide information on respite care.[7] A key finding was the value of the PNS in hospital and community settings, but further clarification of the PNS role in these settings was required. A Swedish study demonstrated the PNSs role in providing tailored and competent care to alleviate the impact of PD on daily life. For the PNS to be effective, they required practical skills, the ability to provide emotional support and needed theoretical knowledge of PD.[13] Further evidence is required about which aspects of the PNSs provide the greatest benefit and the cost-effectiveness of different models of care. To date, there has been no evaluations of the role of the Neurology Nurse and the impact these nurses have on PwP.

### Gaps in the evidence

For the purpose of this project, the term specialist nurse will be used from here on in, and refers to all specialist nurses working in what was traditionally a PNS role. There are significant challenges in collecting evidence about how PNSs achieve improvements in outcomes for PwP due to a plethora of issues:

▶ There are multiple configurations of specialist nurses in the field of Parkinson's with different roles and job titles, across and within different settings.
▶ The scope of different service models in which specialist nurses operate is not understood.
▶ The PNS competency document was published in 2006 but relied heavily on one research study and the knowledge, skills and competencies of the current specialist nurse community is unknown.
▶ The value of the specialist nurse during each stage of PD is unclear.

► Current facilitators and barriers to providing an effective specialist nurse service are poorly understood.

Furthermore, there are significant challenges in identifying the outcome measures to evaluate the specialist nurse's role:

► Specialist nurses often work within MPTs, with different service models; attributing outcomes to the specialist nurse alone is challenging.
► Specialist nurses cover most of the UK, making it neither possible nor ethical to establish a matched control group of PwP without access to a specialist nurse, against which to compare outcomes.
► PD is a progressive condition, with all PwP showing levels of deterioration over time; using health outcomes is not a reliable or effective way of evaluating the value of specialist nurses.

## REALIST EVALUATION

Realist evaluation is a theory driven approach[14] used to further understand complex interventions. The different specialist nurse models used throughout the UK can be thought of as 'complex interventions', as they involve several interacting components that are sensitive to context.[15] Realist approaches explain how and why programmes, policies and interventions work (or not) through striving to answer the question: what works, for whom and under what circumstances.[14 16] In contemporary theory-driven evaluation, theories about a programme are developed in many different ways and are used for a variety of purposes.[17 18] Realist evaluation proposes that for an intervention to work, resources (mechanism) must influence the reasoning (mechanism) of the targeted actors to cause them to adopt an intended behaviour, that in a specific context will lead to a specific outcome.[19] Outcome patterns are found within most social programmes[20] and realist evaluation focuses on exploring these observed differences. This is done by identifying and testing programme theories in the form of Context-Mechanism-Outcome configurations (CMOCs). CMOCs are developed using the formula: intervention resources (M) are introduced in a context (C), in a way that enhances a change in reasoning (M). This alters the behaviour of participants, which leads to outcomes (O).[21] By iteratively developing, refining and testing programme theories, we will be able to understand how, why, for whom and in what circumstances specialist nurses produce desired and undesired outcomes for PwP and their families and carers.

### Evaluation questions and objectives

How do specialist nurses work in the field of Parkinson's? For whom and in what circumstances do they work best?

The research questions above are broad to allow the formulation of underlying assumptions about how specialist nurses work, and what impacts they are expected to have. This is a crucial starting point for realist evaluation and provides the basis for theory development.[14]

The objectives of this study are:

1. To develop and refine realist explanatory theories to understand better the underpinning mechanisms and facilitative contexts that lead to positive and negative outcomes when a specialist nurse is involved in the care of PwP.
2. To ascertain the skills, knowledge, experience and job specification of the specialist nurse.
3. To determine the level of key competencies (as measured by the PNS Competency Framework) within specialist nurses.
4. To explore the specialist nurse role in the trajectory of PD.
5. To explore and gain an understanding of the impact of the different models of specialist nursing on care.
6. To share good practice across the specialist nurse community.
7. To identify, measure and value the resource (cost) implications and outcomes (benefits) linked to the realist programme theories in order to provide explanations of the comparative cost–benefit of different models of specialist nursing.
8. To understand the perceived value and net financial impacts of specialist nurse interventions and make a robust case for commissioning.

## METHODS AND ANALYSIS

We have used the RAMESES II (Realist And MEta-narrative Evidence Syntheses: Evolving Standards) reporting standards for realist evaluations[22] to structure our reporting of the study details, methods and analysis.

### Rationale for using a realist economic evaluation

Realist evaluation is increasingly used to evaluate complex health system interventions because of its appealing practical framework for making sense of them.[23] However, to be successfully and sustainably adopted, policymakers, service managers and practitioners want public programmes to be affordable as well as effective.[24] This 2-year study uses a mixed-methods realist framework incorporating an economic analysis to identify, measure and value the resource implications linked to the realist programme theories about the specialist nurse, as well as value the outcomes. Though exploratory in its endeavour at present, we will attempt to use this approach, herein referred to as 'realist economic evaluation', to provide explanations of the comparative cost-benefits of different models of specialist nursing. We have therefore chosen to adopt a realist economic evaluation design because of its unique ability to address questions of what works, for whom and in what circumstances, while also exploring resource use, costs and cost-effectiveness.[24]

### Design

This research uses a realist evaluation and aims to integrate an economic analysis within the realist framework. We refer to this as 'realist economic evaluation'. In order to facilitate this, the study will comprise four phases. First, we will develop resource-sensitive initial programme

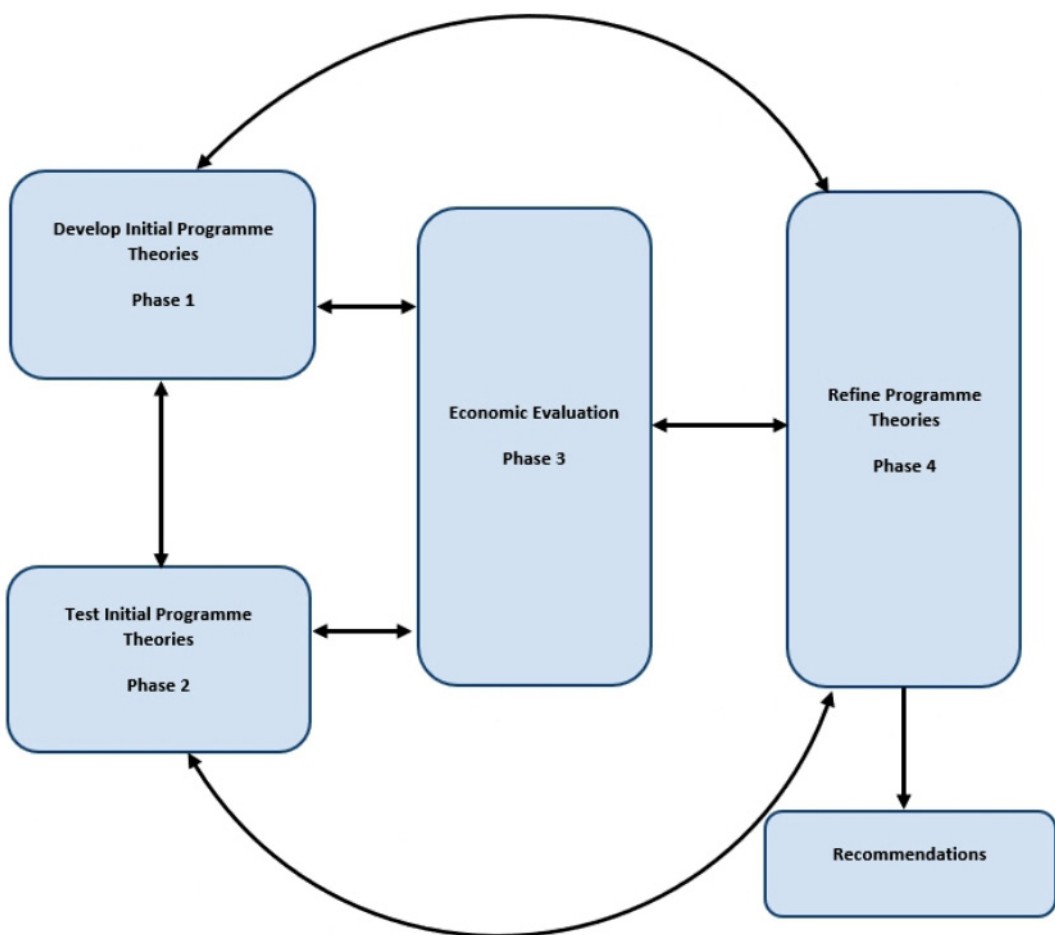

**Figure 1**  Operationalisation of a four-phase realist economic evaluation.

theories (IPTs) (herein referred to as economic informed IPTs) about how specialist nurses 'work' (including associated costs and benefits) in the field of Parkinson's, using surveys with specialist nurses, PwP and their families/carers, to capture similarities and differences in specialist nurses and models of delivery nationally (phase 1). Once we understand the different service models, quantitative analysis of the survey data and qualitative interviews will be conducted with a sample of specialist nurses, PwP and their family members/carers to further explore responses to the survey questions and refine the programme theories (phase 2). The identification and measurement of costs and benefits associated with the programme theories will be integrated and undertaken iteratively within the realist evaluation during phases 1 and 2, as the economic informed IPTs are developed and tested. We will also draw on secondary data sources to value the costs and benefits where appropriate. These economic informed programme theories will be used to compare the different specialist nursing models with respect to their effectiveness, costs and benefits (phase 3). The findings from all three phases will be integrated to iteratively refine the programme theories (phase 4). This unique design will provide a comprehensive and detailed realist economic evaluation. The four-phase study design

can be seen in figure 1, although consistent with a realist approach, operationalisation of the study design is an iterative process.

The study has been developed and will be overseen by an expert panel including: the research team, Parkinson's Disease Nurse Specialist Association (PDNSA) representatives, nurse leads within the Parkinson's Excellence Network, the Parkinson's UK Service Development and Improvement Lead, three PwP and a carer.

### Developing potential IPTs

An example of an IPT, which includes 'financial realist resources',[24] is provided below (box 1). IPTs developed as part of the initial funding application, were structured in

---

**Box 1  Initial programme theory example**

► Initial programme theory: Due to PNS's advanced training (context) they have the skills and feel confident to deliver information on self-management to PwP (resource); PwP understand the strategies and have increased confidence to self-manage (reasoning) which leads to a reduction in seeking care and fewer interactions with health-care professionals (outcomes).

PNS, Parkinson's nurse specialists; PwP, people with Parkinson's.

---

line with the Parkinson's Nurse Competency Framework[11] which covers: counselling, access, monitoring, symptoms management, and research and development.

## Data collection methods

### Phase 1: Developing the initial programme theories

The expert panel reviewed previous PNS surveys and developed questions for a new specialist nurse survey. They also discussed potential links between resources, including economic resources (such as service and patient costs) and outcomes (such as service use, quality of life and well-being). Initial theories and CMOCs were therefore expanded to include theories around economic resource use (costs) and outcomes (benefits), we refer to these as economic informed IPTs. These economic informed IPTs were used to develop the questions in the surveys. In addition, PPI team members developed questions for two separate surveys: one for PwP and one for family members/carers of PwP. These initial engagements were informal and guided by the stakeholders to ensure we captured data relevant and useful to real-life settings.

The purpose of the nurse survey is to capture the models of delivery of specialist nurses working in the field of Parkinson's. It will explore areas such as role, working practices, caseload, qualifications, experience and level of competence and will thus provide insight into the contexts, structures and strategies around specialist nurse interventions. Although we are specifically targeting Parkinson's nurses, the survey will be open to any specialist practitioner (eg, neurology nurses, physiotherapists, occupational therapists) whose day-to-day role involves caring for people with Parkinson's. We anticipate over 400 responses. This will provide us with a large amount of quantitative data that we can compare and contrast and it will also help us to define the different service models of specialist nursing. We will recruit over a period of 6 months via Parkinson's UK, local CRNs and social media. The PDNSA and National Institute for Health Research will also provide support with the identification of potential participants, particularly those who are not PNSs but are fulfilling that role. The survey will be promoted by the PDNSA, the expert panel group, the Parkinson's UK Excellence Network and local PNS groups. All study information and a link to the survey will be shared on a Northumbria University website, which potential participants will be directed to.

The purpose of the surveys for PwP and carers is to gain a better understanding of the role of the specialist nurse and the impact they have on the lives of PwP and their carers, thus providing insight into the mechanisms triggered by specialist nurse interventions and the associated outcomes. The surveys will be open to any person diagnosed with idiopathic Parkinson's disease and any family member, friend or informal carer of a person with idiopathic Parkinson's disease. The surveys for PwP and families/carers will explore areas such as access to a specialist nurse (ie, when, where, for how long), information and support provided by a specialist nurse, and perceived benefits, limitations and impact of their specialist nurse on their health and well-being (collected using validated quality of life, satisfaction questionnaires[25 26] and service use measures). We will recruit over a period of 6 months via Parkinson's UK, specialist nurses, local CRNs and social media channels, using the study's website link.

At the end of this phase, the information collected will provide the research team with data to finalise the economic informed PTs, using retroduction to infer relationships between items investigated in the surveys. This is referred to as theory gleaning through survey work.[27] For example, if the nurse survey indicates that there are more telephone services available throughout the UK and the PwP survey indicates that this is preferential, theorising (using the survey results) will take place around why this is preferential, how and for who, including theorising around the additional costs (positive and negative i.e. cost savings) and benefits (positive and negative) of providing the service. All survey participants will be asked if they would be happy to take part in a semi-structured interview to further explore the role of the specialist nurse and the issues identified in the survey. Those who select 'yes' will be invited to provide their email address so that they can be contacted directly by the research team.

### Phase 2: Testing the initial programme theories

Following the specialist nurse survey, up to 15 specialist nurses will be invited to participate in realist interviews to explore their perceptions of their role and activities. Participants will be purposively selected to ensure a cross-section of the different specialist nurses is represented (ie, community-based PNS, hospital-based PNS and other specialist nurses working in the field of Parkinson's).

Realist interviews will also be conducted with up to 15 PwP and 10 family members/carers. Participants will be purposively selected to ensure comparisons can be made across the different models of specialist nursing. Only those PwP with a diagnosis of idiopathic PD according to the UK Brain Bank Criteria,[28] with capacity and ability to give written informed consent, will be recruited. PwP with no access to a specialist nurse will be able to complete the survey but will be directed to appropriate questions, and will be given the opportunity to take part in an interview.

Data from the surveys and semi-structured interviews will allow us to examine the impact of the specialist nurse and test the economic informed programme theories developed in phase I.

### Phase 3: Economic evaluation

To date, economic evaluations in PD have compared nurse vs GP or consultant-led services.[5 8] It could be argued that in order to harness the benefits of multiprofessional working, both professional groups are needed to work collaboratively in order to provide quality of care. As such, a more informative evaluation would be to determine the most efficient service delivery model of the specialist nurse.

Drawing on and expanding realist economic evaluation principles laid out by Anderson and Hardwick,[24] we will employ a cost–benefit analysis to compare the resource requirements (costs) and value the outcomes (benefits including the wider, social benefits beyond mental and physical well-being) associated with programme theories that underpin the different specialist nurse models. As described above, economic informed programme theories, will be iteratively developed through phase 1 (IPT development) and phase 2 (testing programme theories). At the end of phase 1, we will have a set of programme theories about the different models of specialist nurse provision that also identify the health economic implications in terms of resource use (such as service and patient costs) and outcomes (such as service use, quality of life and well-being). At the end of phase 2, we will have tested these programme theories using resources use and outcome data to provide a greater understanding of the relationship between context, mechanism and health economic outcomes (costs and benefits). It is not possible at this stage to state exactly what resource and outcome data will be required for the analysis until we understand the resource use and outcome implications that will evolve within the economic informed realist programme theories. As an example, they will likely include measures of staff training, time, travel, referrals, equipment, social support, patient and carer resource use, OOH care, 999/111 ambulance calls, A&E attendances, non-elective admissions, GP visits, telephone consultations, symptoms management, quality and life and well-being measures. These data will be gathered prospectively in the surveys and interviews where possible (eg, we have included health service utilisation measures in the surveys), and also retrospectively using secondary data such as the Excellence Network Data Dashboards,[29] to evaluate the differences and trends in hospital admissions. Resource use will be costed at standard national tariffs.

By the end of phase 4 (see below), the refined economic informed programme theories will be used to build a comparative model the effectiveness, costs and benefits of the different services of specialist nurse provision. While this is the form we expect the realist economic evaluation to take, this is ground-breaking new research and therefore we will employ an explorative approach; maintaining flexibility with the methods and being driven by programme theory, in line with the realist approach.

### Phase 4: Refining the programme theories

Although theory refinement is presented as a separate phase, it will take place throughout the duration of the project as per the iterative nature of realist evaluation. This will be done via team meetings where we will think about and discuss the programme theories individually and what the data adds to them.

### Data analysis

The project will use an overarching realist framework and qualitative data will be analysed using a realist logic of analysis.[14] All interviews will be audio-recorded (subject to informed consent) and transcribed verbatim. The transcriptions will be coded using the NVivo qualitative data analysis software to allow refinement of the programme theories. This will be done by coding data under the following headings: context, mechanism, outcome, potential CMOC, supports/refutes/refines, how/why/decision-making processes, links to other IPTs and additional notes.[30] As new questions emerge and connections are established, the literature will be revisited, thus deepening understanding and meaning of the findings.[31] This iterative and reflective process will serve as a tool for analysis and will allow greater transparency for how and why the CMOCs are developed and refined.[30]

Quantitative data analysis will be supported by the statistical software package IBM SPSS (IBM, Armonk, New York, USA) and will be overseen by an experienced statistician. Data will be summarised using appropriate summary statistics (eg, mean, median, proportion), depending on the level and nature of the data (eg, parametric, non-parametric, ordinal, frequency). Likewise, inferential analysis (eg, t-tests, Mann-Whiney U tests, $\chi^2$ tests) will be used as appropriate to the data to complement, inform and support qualitative analysis.

The programme theories will draw on both quantitative and qualitative data, with quantitative data often consisting of outcomes and qualitative data used to explain associated contexts and mechanisms.[14] The data collection methods are, therefore, thoroughly integrated. By integrating the findings across the four phases, we will be able to refine the programme theories and explain how specialist nurses work in the field of Parkinson's in the UK and what impact they have on PwP and their families and carers.

### PATIENT AND PUBLIC INVOLVEMENT

PPI team members have been involved in this research from the outset. In an initial meeting in May 2019, three PwP and a carer volunteered to be actively involved in the design, management, recruitment and conduct of this study. They recruited three more PwP/family members/carers for survey development consultations, which took place in July and October 2019. This additional PPI recruitment enables the burden of work to be shared and ensures consistent PPI input in the unfortunate case of illness. The discussions of the consultations enabled us to develop questions for the PwP and carer surveys and supported IPT development. PPI members have also been involved in promoting the study via social media; the study website features information about them and they are co-authors of this protocol.

### ETHICS AND DISSEMINATION
#### Ethical approval

This research was approved through Northumbria University's Ethical Approval System (reference number:

17683) and the Health Research Authority and Health and Care Research Wales (REC reference number: 19/EE/0254).

## Dissemination of findings

Key findings will be disseminated via the study website throughout the duration of the project. We have an active Twitter account for the study and we will continuously post updates and direct people to the website. Parkinson's UK and the Excellence Network will support promotion of the study and dissemination of the findings throughout the project via direct emails to mailing lists, annual and regional nurse meetings, conferences, newsletters, online news pages and an online 'Take Part Hub'. Learnings and improvements in service delivery approaches will be implemented when the information becomes available, rather than after the full research cycle has been completed. Such an approach will make policy and practice-relevant research immediately available to end users, accelerating the use of evidence in decision-making of health and other services.[32] Academic dissemination will occur through publication and conference presentations.

**Contributors** AH conceived the study and developed the study protocol. SB led the writing of the manuscript with input from SD, AB, AH, KB, AC, RB and CA. SMD and AB developed the realist economic evaluation component of the methods. All authors reviewed and critiqued the manuscript and approved the final published version.

**Funding** This study is funded by Parkinson's UK (Grant reference: G-1807) and the sponsor is Northumbria Healthcare NHS Foundation Trust.

**Disclaimer** The views expressed in this paper do not necessarily represent those of the funders. The funders had no role in study design, data collection and analysis, decision to publish, or preparation of the manuscript.

**Competing interests** None declared.

**Patient and public involvement** Patients and/or the public were involved in the design, or conduct, or reporting, or dissemination plans of this research. Refer to the Methods and analysis section for further details.

**Patient consent for publication** Not required.

**Provenance and peer review** Not commissioned; externally peer reviewed.

**ORCID iDs**
Sarah Brown http://orcid.org/0000-0002-7078-0984
Sonia Michelle Dalkin http://orcid.org/0000-0002-3266-5926
Annette Hand http://orcid.org/0000-0002-9364-757X

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
