## [Reviewer comments · BMJ Open]

ARTICLE DETAILS

TITLE (PROVISIONAL)	Exploring and understanding the scope and value of the Parkinson's nurse in the UK (The USP Project): a realist economic evaluation protocol
AUTHORS	Brown, Sarah; Dalkin, Sonia; Bate, Angela; Bradford, Russ; Allen, Charlotte; Brittain, Katie; Clarke, Amanda; Hand, Annette

VERSION 1 - REVIEW

REVIEWER	Anne Wissendorff Ekdahl Lunds University
REVIEW RETURNED	02-May-2020

GENERAL COMMENTS	A very interesting project with the aim of studying/evaluating nurses' way of working with people with Parkinsons' Disease.
---

REVIEWER	Marguerite Bramble Charles Sturt University Australia
REVIEW RETURNED	04-May-2020

GENERAL COMMENTS	Exploring and understanding the scope and value of the Parkinson's nurse in the UK (The USP Project): a realist economic evaluation protocol • This is a well-constructed and designed study protocol using Anderson's untested analysis however I would like to see the Economic Evaluation commence before Phase 3 and reflected in more detail in the study design as follows:a. Include in Phase 1 a Program Logic or other mechanisms-contexts-outcomes configuration as a basis for exploring the framework for cost-effectiveness comparison analysis across programmes.b. Identify more clearly in Phase 2 the relationship between cost-effectiveness and programme theories in terms of resource (cost) implications and outcomes (benefits).c. Costing data is then available for the economic evaluation in Phase 3 and will produce more explanations for refining Programme Theories in Phase 4.Otherwise I have difficulty with the term 'Realist economic evaluation protocol'. As it is the economic evaluation is 'nested' in the study
--

	protocol as an afterthought and insufficient data will be collected to understand 'what works'.  Dates of study are not included in the manuscript.
--	---

REVIEWER	Roman Ayele University of Colorado-School of Public Health, Denver, Colorado
REVIEW RETURNED	16-May-2020

GENERAL COMMENTS	Great protocol paper describing their project on the role of PNS on PD patient outcomes. Well described qualitative and quantitative approaches.
--

VERSION 1 – AUTHOR RESPONSE

Reviewer: 1

We would like to thank reviewer 1 for their time, there were no comments to respond to.

Reviewer: 2

We would like to thank reviewer 2 for their time and constructive comments. Our responses to their comments are given below.

R2: This is a well-constructed and designed study protocol using Anderson's untested analysis however I would like to see the Economic Evaluation commence before Phase 3 and reflected in more detail in the study design as follows:

a. Include in Phase 1 a Program Logic or other mechanisms-contexts-outcomes configuration as a basis for exploring the framework for cost-effectiveness comparison analysis across programmes. Thank you, this was not clear so we have added a paragraph in Phase 1 detailing how health economic concepts of cost and benefit were embedded in the development of the initial programme theories and that CMO configurations were extended to include these concepts. We have also added to the IPT example that was given in this Phase to illustrate this.

b. Identify more clearly in Phase 2 the relationship between cost-effectiveness and programme theories in terms of resource (cost) implications and outcomes (benefits). Again, thank you, we appreciate that this was not clear. However, we believe that further clarity on this is better explained in Phase 3 so we have expanded this section to include further detail on how explanations of resource use and outcomes will be embedded in the development of programme theory and how programme theories will be tested and refined using primary data from the surveys and interviews, and secondary data if required, as well as, how these will inform a comparative analysis of the costs and benefits of the specialist nurse models. In addition, we have expanded the design section to make clear the links between the development and refinement IPTs including the cost and benefit implications, and the comparative analysis of the effectiveness, costs and benefits of the specialist nurse models.

c. Costing data is then available for the economic evaluation in Phase 3 and will produce more explanations for refining Programme Theories in Phase 4.

Otherwise I have difficulty with the term 'Realist economic evaluation protocol'. As it is the economic evaluation is 'nested' in the study protocol as an afterthought and insufficient data will be collected to understand 'what works'.

Thank you for these comments. As above, we appreciate this was not clear and have expanded on these sections to make it clear that economic concepts of cost and benefit are central to development and testing of the programme theories from Phase 1, indeed we have reworded these as economic informed programme theories. As such the data required for testing the theories will be gathered as we move iteratively through Phases 1-4. The confusion is in part due to the way the manuscript is presented as it guides the reader to move through the phases sequentially – making it look like the economic analysis (phase 3) is nested within the realist evaluation – however, the phases are designed to be conducted iteratively with each phase informing and being informed by one another. This is further illustrated in figure 1. As such the economic evaluation is embedded within the realist evaluation from the start of Phase 1.

Dates of study are not included in the manuscript.

Thank you, study dates have now been added to the manuscript.

Reviewer: 3

We would like to thank reviewer 3 for their time, there were no comments to respond to.

VERSION 2 – REVIEW

REVIEWER	Marguerite Bramble Charles Sturt University Australia
REVIEW RETURNED	09-Jul-2020

GENERAL COMMENTS	Overall I am satisfied with the additions to Phase 1 and 2 of the study design with the aim to integrate the economic analysis as part of the realist evaluation. I like the way you have expanded on the phases by referring to ‘economic informed IPTs’ to gain a better understanding at the end of Phase 2 of the relationship between context, mechanism, and health economic outcomes using the CMOC framework. I think the modified study design diagram now reflects the four phases. I have noted one very small edit below in CAPS. I wish you all the best with this very interesting and valuable study. This manuscript is ready for publication. Additional edits 1. By the end of Phase 4 (see below), the refined economic informed programme theories will be used to build a comparative model OF the effectiveness, costs and benefits of the different services of specialist nurse provision.
---